# Design, Synthesis, and Biological Evaluation of HDAC Degraders with CRBN E3 Ligase Ligands

**DOI:** 10.3390/molecules26237241

**Published:** 2021-11-29

**Authors:** Yingxin Lu, Danwen Sun, Donghuai Xiao, Yingying Shao, Mingbo Su, Yubo Zhou, Jia Li, Shulei Zhu, Wei Lu

**Affiliations:** 1Shanghai Engineering Research Center of Molecular Therapeutics and New Drug Development, School of Chemistry and Molecular Engineering, East China Normal University, 3663 North Zhongshan Road, Shanghai 200062, China; luyingxin_roy@163.com (Y.L.); cpu_zsl@163.com (D.X.); 2National Center for Drug Screening, State Key Laboratory of Drug Research, Shanghai Institute of Materia Medica, Chinese Academy of Sciences, 189 Guo Shoujing Road, Shanghai 201203, China; danwensun1@163.com (D.S.); yingying_shao1226@163.com (Y.S.); mbsu@simm.ac.cn (M.S.); ybzhou@simm.ac.cn (Y.Z.)

**Keywords:** HDAC degraders, CRBN ligands, proteins of interest, benzyl alcohol linkage

## Abstract

Histone deacetylases (HDACs) play important roles in cell growth, cell differentiation, cell apoptosis, and many other cellular processes. The inhibition of different classes of HDACs has been shown to be closely related to the therapy of cancers and other diseases. In this study, a series of novel CRBN-recruiting HDAC PROTACs were designed and synthesized by linking hydroxamic acid and benzamide with lenalidomide, pomalidomide, and CC-220 through linkers of different lengths and types. One of these PROTACs, denoted **21a**, with a new benzyl alcohol linker, exhibited comparably excellent HDAC inhibition activity on different HDAC classes, acceptable degradative activity, and even better in vitro anti-proliferative activities on the MM.1S cell line compared with SAHA. Moreover, we report for the first time the benzyl alcohol linker, which could also offer the potential to be used to develop more types of potent PROTACs for targeting more proteins of interest (POI).

## 1. Introduction

Histone deacetylases (HDACs) have been identified as a superfamily of enzymes serving as “epigenetic erasers” [1,2]. They play important roles in altering the gene transcription and drug resistance mechanisms of tumor cells due to their ability to remove the acetyl moiety from histone tails [3,4]. Eighteen HDACs are classified into two families based on their catalytic mechanisms, including eleven zinc-dependent metalloenzymes HDAC1-11, which are grouped into three classes based on their homology: class I (HDACs 1, 2, 3, and 8), class II (HDACs 4, 5, 6, 7, 9, and 10), and class IV (HDAC 11); and seven nicotinamide adenine dinucleotide (NAD^+^)-dependent sirtuins SIRT1-7, grouped into class III [5]. Among these, zinc-dependent metalloenzymes are well-known for their importance in cell motility, immunoregulation, and proliferation [6].

In the past few decades, hundreds of HDAC inhibitors (HDACi) have been developed for the treatment of many malignant and non-malignant diseases [7,8]. Typical HDACis can be categorized into four subtypes, based on their chemical structure: (1) short-chain fatty acid; (2) hydroxamic acid; (3) benzamides; and (4) cyclic peptides [9,10,11]. Despite its structural distinctiveness, HDACi is generally considered to share a common key motif, a “war-head”, such as hydroxamic acid, that interacts with the Zn^+^ in the active site, which is crucial for histone deacetylation activity. To date, hundreds of hydroxamic acid- and benzamide-type HDACi have been submitted to clinical trials (Figure 1), such as vorinostat (SAHA), panobinostat, belinostat, and chidamide [12,13,14]. Unfortunately, these drugs exhibit limited selectivity, which has been attributed as a cause of their debilitating side effects in patients [12,15]. Hence, it is urgent to develop an approach to enhance selective HDAC binding.

Recently, a new technology, the proteolysis-targeting chimera (PROTAC) strategy, has attracted a great deal of attention in drug discovery due to its unique mechanism, which was first proposed by Crews and Deshaies in 2001 [16,17,18,19,20]. PROTACs are heterobifunctional molecules consisting of three parts: an E3 ligase system recruiting ligand acting as an “anchor”; a protein of interest (POI) binding moiety acting as a “warhead”; and a chemical linker acting as a “bridge” [21]. Through binding to both proteins in cells, a POI is recruited by PROTACs to a ternary complex (TC) with the E3 ligase, which triggers the ubiquitin-proteasome system (UPS), causing the degradation of the targeted protein by 26S proteasome [22,23]. Recently, different completely new classes of small-molecule therapeutic agents PROTACs have been discovered and developed with cereblon (CRBN) and von Hippel-Lindau (VHL) as E3 ligases (Figure 2). These have attracted much attention [24,25]. Thus, employing PROTAC technology in HDAC degradation is a promising technology in drug discovery.

C-220 (Figure 3), has been designed as the cereblon modulator, which is currently in clinical trials for the treatment of systematic lupus erythematosus (SLE) and relapsed and refractory multiple myeloma (MM) [26,27]. CC-220 has been demonstrated to bind to CRBN through the glutarimide moiety, same as lenalidomide and pomalidomide, and to feature a higher affinity for CRBN than these moieties [28]. Typically, most of the PROTACs with lenalidomide and pomalidomide as CRBN ligands were linked with the “warhead” in the four-amino position with an alkyl linker or a polyethylene glycol linker [29,30]. To the best of our knowledge, no degraders of HDACs with benzyl alcohol as a linker have been reported, and whether this type of PROTAC can promote degradation remains to be determined (Figure 3).

Based on the above reasons, accordingly, we designed and synthesized a series of novel CRBN-recruiting HDAC PROTACs by linking hydroxamic acid and benzamide with lenalidomide and pomalidomide through linkers of different lengths and types (Figure 3). Further, we report for the first time the benzyl alcohol linkage between the four-position of lenalidomide or pomalidomide and the “warhead”, which can be used in other types of PROTACs (Figure 3C). The resulting products were further evaluated in HDAC inhibition studies, in vitro anti-proliferative studies, and western blot assay; profiling in these cell-based assays revealed that our bifunctional small molecules selectively promoted the degradation of HDAC through the UPS.

## 2. Results and Discussion

### 2.1. Chemistry

The synthesis of the presented final compounds was outlined in Figure 1, Figure 2 and Figure 3. The synthetic routes towards **9a**–**9e** from the commercially available compounds **1a**–**1d** are outlined in Figure 1. The synthesis of the intermediates **2a**–**2d** and the linkers **4a**–**4d** followed published procedures [31,32,33,34]. Next, with lenalidomide (**5a**) and pomalidomide (**5b**) as the CRBN ligands, the key intermediates (**6a**–**6e**) with varying linker lengths were obtained through an amide reaction followed by the deprotection of the benzyl group using palladium on carbon (Pd/C) in a hydrogen atmosphere with high yields. Subsequently, a condensation reaction was applied to couple **7a**–**7e** with triphenylmethyl (Trt) hydroxylamine, using HATU as a condensation reagent, to obtain **8a**–**8e** in yields between 62 and 79%. The subsequent Trt-deprotection of compounds **8a**–**8d** and **8e** created the corresponding lenalidomide-conjugated compounds **9a**–**9d** and pomalidomide-conjugated compound **9e**.

As shown in Figure 2, the synthesis of compounds **15a**–**15d** commenced with the conversion of lenalidomide (**5a**) and pomalidomide (**5b**) to carbamates **11a** and **11b**, respectively, using four-nitrophenyl chloroformate in THF. The reaction of **11a** and **11b** with amino acids (**12a**–**12c**) with varying lengths under mild conditions created the carboxylic acid **13a**–**13d**, which were subsequently reacted with triphenylmethyl (Trt) hydroxylamine to create **14a**–**14d** in acceptable yields. Similarly, **14a**–**14d** also underwent mild acidic deprotection to create the corresponding lenalidomide-conjugated compounds **15a**–**15c** and pomalidomide-conjugated compound **15d**.

As shown in Figure 3, **21a**–**21b** and **23a**–**23b** were synthesized according to the following procedure. Using lenalidomide-4-OH (16a) and pomalidomide-4-OH (**16b**) as starting materials and CRBN ligands, the compounds **19a**–**19b** were obtained through a substitution reaction resulting in **18a**–**18b**, followed by removal of the Boc-protecting group under acidic conditions. Employing compounds **19a**–**19b** as intermediates, the final compounds **21a**–**21b** were synthesized by condensation with o-phenylenediamine (**20**) and **23a**–**23b** were synthesized by condensation with O-tritylhydroxylamine followed by the subsequent removal of the Trt group.

### 2.2. Discussion and Biological Evaluation

Because the length and type of the linker plays a key role in the potency of PROTAC degrader molecules, **9a**–**9e** and **15a**–**15d**, with varying linker lengths, were synthesized through amide linkage and urea linkage, respectively. The molecules **21a**–**21b** and **23a**–**23b** were also synthesized with benzyl alcohol linkage under moderate conditions. It should be noticed that inhibition does not always correlate with degradation [35]. The HDAC inhibitory ability of compounds is one of the most critical parameters in HDACi, and HDAC degraders. Thus, the HDAC inhibitory ability of these molecules was first evaluated on several recombinant HDAC enzymes, including HDAC1, HDAC3, and HDAC6, at a concentration of 20 μg/mL, with SAHA as the positive control. As shown in Table 1 and Appendix A, **9b**, **9d**, **9e**, **15a**, **15b**, **15c**, and **15d** exhibited high HDAC1 or HDAC3 inhibitory abilities with a nano-molar level IC_50_ value. Unfortunately, compounds **9a** and **9c** displayed sharply decreased potencies in the inhibition of HDAC1 and HDAC3, suggesting that the inhibitory abilities are related to the linker lengths. The molecules **21a**–**21b** and **23a**–**23b**, with the benzyl alcohol linkage, also displayed similar or slightly weaker inhibitory abilities than SAHA, suggesting their potential use in the development of more types of potent PROTACs. Surprisingly, most of the molecules exhibited much weaker HDAC6 inhibitory abilities compared with the control SAHA, and the mechanism needs further elucidation.

Based on the high HDAC inhibition activities of the molecules, we next investigated the in vitro anti-proliferative activities of the newly synthesized compounds. It was reported previously that MM.1S cells were more sensitive to HDAC PROTAC molecules [1,12]. Consequently, in this study, in vitro anti-proliferative activities of the molecules chosen from the in vitro HDAC inhibition was performed against the MM.1S cell line. As shown in Table 2 and Appendix A, **21a** exhibited the most potent in vitro anti-proliferative activities against the MM.1S cell line with an IC_50_ value of 0.043 μM, which was approximately five times more potent than that of SAHA. Meanwhile, 21b also exhibited comparable anti-proliferative activity with SAHA, with an IC_50_ value at a nano-molar level. However, the anti-proliferative activities of **9b**, **9e**, **15a**, and **15d** were significantly weaker than those of **21a**, **21b**, and **23b**. A possible explanation for this phenomenon may be that degraders with a shorter linker could induce the degradation of HDAC and further increase the anti-proliferative activities, which deserves further immunoblot studies. Interestingly, the HDAC inhibitory abilities of **9b**, **9e**, **15a**, and **15d** were significantly stronger than those of **21a**, **21b**, and **23b**. The discrepancy between IC50s in the inhibition of HDAC6 and the anti-proliferation for compound **21a** strongly suggests that the anti-proliferation of **21a** operates under a very different mechanism.

Subsequently, to evaluate the degradation potential of the molecules, we treated the MM.1S cells with **21a**, **21b**, and **23b** with different concentration gradients based on the previous in vitro anti-proliferative activities for 3 h, and checked the HDAC1, HDAC3 and HDAC6 protein levels by immunoblotting. As illustrated in Figure 4A, the amount of HDAC proteins remained the same when the cells were treated with different concentrations of **21b** and **23b**, respectively, suggesting almost no degradation of HDAC1, HDAC3, or HDAC6. These results were significantly different from the anti-proliferative results, indicating that **21b** and **23b** function through an enzyme inhibition mechanism rather than through degradation. Among the three tested molecules, degrader **21a** demonstrated superior HDAC6 degradation efficacies at a concentration of 0.12 μM to HDAC1 and HDAC3. The selective degradation of HDAC6 over HDAC1 and HDAC3 was unexpected, but not surprising, because it has been reported that selective degraders can be developed by tethering non-selective BRD ligands with an E3 ubiquitin ligase ligand. Compared with the HDAC inhibitory ability results, the HDAC6 degradation ability suggested that **21a** functions through a degradation mechanism.

To further investigate the mechanism of the action of degrader **21**, the effective concentration range and dependence of **21a** towards HDAC6 degradation was conducted in the MM.1S cell line, with SAHA as a control. As shown in Figure 4B, the degradation of HDAC6 by **21a** was observed in a dose-dependent manner. However, in the present study, a notable increase in tubulin acetylation was not observed and the degradation potency of **21a** still needs to be improved in further study. Overall, these results indicated that **21a** offers the potential to selectively degrade HDAC6 and suggested the possibility of using benzyl alcohol linkage to develop more types of potent PROTACs, which deserves further evaluation.

## 3. Conclusions

In this study, a series of HDACi-based PROTACs were designed and synthesized by linking hydroxamic acid and benzamide with lenalidomide, pomalidomide, and CC-220 through linkers of different lengths and types as a novel class of HDAC degraders. Overall, with the synthesis of these heterobifunctional molecules, we have demonstrated the feasibility of the targeted degradation of HDAC. It seems likely that the linker lengths and types of HDAC degraders will exert a profound effect on their HDAC and anti-proliferative activity in cells. Among these degraders, **21a**, with a new benzyl alcohol linker exhibited comparably excellent HDAC inhibition activity on different HDAC classes and degradative activity on the MM.1S cell line compared with SAHA. Furthermore, **21a** also exhibited even better in vitro anti-proliferative activities on the MM.1S cell line, demonstrating great therapeutic potential for the treatment of certain disorders through HDAC degradation. Taken together, these results demonstrate important parameters for PROTAC design and reveal that further optimization is still needed to develop this novel type of HDAC technology. Moreover, we report for the first time the benzyl alcohol linkage, which could also offer the potential to be used to develop more types of potent PROTACs for targeting more POIs.

## 4. Materials and Methods

### 4.1. Materials

All the reagents were obtained from commercial sources and dried prior to use, unless otherwise stated. Anhydrous solutions of reaction mixtures were transferred via an oven-dried syringe or cannula. All the reactions were monitored by thin-layer chromatography (TLC) on 25.4 mm × 76.2 mm silica 394 gel plates GF-254 and UPLC-Mass on Waters ACQUITY UPLC H-Class. The ^1^H NMR and ^13^C NMR spectra were recorded at 23 °C in CDCl_3_ and DMSO-*d_6_* on a Bruker DRX-400 (400 MHz) using TMS as the internal standard. Chemical shifts were reported as δ (ppm) and signal splitting patterns were described as singlet (s), doublet (d), triplet (t), quartet (q), quintet (quint), or multiplet (m), with coupling constants (*J*) in hertz. High-resolution mass spectra (HRMS) were obtained on an electron spray injection (ESI) Thermo Fisher Scientific LTQ FTICR mass spectrometer. The purity of all the tested compounds was ≥ 95%, as estimated by HPLC analysis performed on Agilent Diamonsil C18 (250 mm × 4.6 mm).

### 4.2. Abbreviations

POI, protein of interest; TC, ternary complex; UPS, ubiquitin-proteasome system; VHL, von Hippel-Lindau; CRBN, cereblon; HDAC, histone deacetylase; DMF, *N*,*N*′-Dimethylformamide; HATU, O-(7-Azabenzotriazol-1-yl)-*N*,*N*,*N*′,*N*′-tetramethyl uranium hexafluorophosphate; DIEA, *N*,*N*-diisopropylethylamine; DCM, dichloromethane; TFA, trifluoroacetic acid; THF, tetrahydrofuran.

### 4.3. Experimental Procedures

Compounds **2a**–**2d** were synthesized according to literature reports [31,32].

Oxocane-2,8-dione (**2a**) (white solid, yield 83%): ^1^H NMR (400 MHz, CDCl_3_) δ 2.40 (t, *J* = 7.2 Hz, 4H), 1.62 (t, *J* = 7.4 Hz, 4H), 1.43–1.29 (m, 2H).

Oxonane-2,9-dione (**2b**) (white solid, yield 76%): ^1^H NMR (400 MHz, CDCl_3_) δ 2.43 (t, *J* = 7.2 Hz, 4H), 1.58 (t, *J* = 7.2 Hz, 4H), 1.40–1.31 (m, 4H).

Oxonane-2,10-dione (**2c**) (white solid, yield 89%): ^1^H NMR (400 MHz, CDCl_3_) δ 2.45 (t, *J* = 7.4 Hz, 4H), 1.68–1.60 (m, 4H), 1.39–1.28 (m, 6H).

Oxonane-2,11-dione (**2d**) (white solid, yield 85%): ^1^H NMR (400 MHz, CDCl_3_) δ 2.46 (t, *J* = 7.2 Hz, 4H), 1.63–1.57 (m, 4H), 1.42–1.23 (m, 8H).

Compounds **4a**–**4d** were synthesized according to previous published reports [33,34].

7-(Benzyloxy)-7-oxoheptanoic acid (**4a**) (white solid, yield 75%): ^1^H NMR (400 MHz, CDCl_3_) δ 7.41–7.24 (m, 2H), 5.11 (s, 2H), 2.35 (dt, *J* = 11.8, 7.5 Hz, 4H), 1.74–1.58 (m, 4H), 1.44–1.26 (m, 2H).

8-(Benzyloxy)-8-oxooctanoic acid (**4b**) (white solid, yield 69%): ^1^H NMR (400 MHz, DMSO-*d*_6_) δ 11.92 (s, 1H), 7.33–7.21 (m, 5H), 5.03 (s, 2H), 2.29 (t, *J* = 6.8 Hz, 2H), 2.14 (t, *J* = 7.2 Hz, 2H), 1.50–1.36 (m, 4H), 1.28–1.13 (m, 4H).

9-(Benzyloxy)-9-oxononanoic acid (**4c**) (white solid, yield 66%): ^1^H NMR (400 MHz, DMSO-*d*_6_) δ 11.97 (s, 1H), 7.40–7.32 (m, 5H), 5.09 (s, 2H), 2.34 (t, *J* = 7.3 Hz, 2H), 2.18 (t, *J* = 7.3 Hz, 2H), 1.59–1.41 (m, 4H), 1.30–1.17 (m, 6H).

10-(Benzyloxy)-10-oxononanoic acid (**4d**) (white solid, yield 64%): ^1^H NMR (400 MHz, DMSO-*d*_6_) δ 12.03–11.91 (m, 1H), 7.46–7.27 (m, 5H), 5.20–4.99 (m, 2H), 2.34 (t, *J* = 7.3 Hz, 2H), 2.18 (t, *J* = 7.3 Hz, 2H), 1.61–1.41 (m, 4H), 1.28–1.20 (m, 8H).

*General procedures for the preparation of compounds **6a**–**6e**.* To a solution of **4a**–**4d** (1.0 eq) in anhydrous DMF, HATU (2.0 eq) and DIEA (3.0 eq) were added. After stirring in an ice bath for 0.5 h, **5a** or **5b** (1.0 eq) was added. The mixture was stirred at room temperature overnight. After the reaction completed, the residue was diluted with H_2_O and extracted with EtOAc. The combined organic layers were then washed with HCl (0.1 M), saturated brine, dried over anhydrous Na_2_SO_4_, and concentrated in vacuo to afford the crude product, which was purified by column chromatography with DCM/ MeOH (30:1–20:1) to produce **6a**–**6e**.

Benzyl 7-((2-(2,6-dioxopiperidin-3-yl)-1-oxoisoindolin-4-yl)amino)-7-oxoheptanoate (**6a**). (white solid, yield 82%): ^1^H NMR (400 MHz, DMSO-*d*_6_) δ 11.01 (s, 1H), 9.76 (s, 1H), 7.81 (d, *J* = 6.8 Hz, 1H), 7.49 (q, *J* = 7.3 Hz, 2H), 7.35 (s, 5H), 5.14 (dd, *J* = 13.2, 4.9 Hz, 1H), 5.08 (s, 2H), 4.45–4.28 (m, 2H), 2.90 (dd, *J* = 21.7, 9.1Hz, 1H), 2.60 (d, *J* = 17.3 Hz, 1H), 2.43–2.27 (m, 5H), 2.08–1.95 (m, 1H), 1.60 (dd, *J* = 15.5, 8.0 Hz, 4H), 1.34 (dd, *J* = 14.8, 7.6 Hz, 2H). ^13^C NMR (100 MHz, DMSO) δ 172.8, 172.7, 171.2, 171.0, 167.8, 136.2, 133.8, 133.7, 132.6, 128.6, 128.4, 127.9, 127.9, 125.2, 119.0, 65.3, 51.5, 46.5, 35.6, 33.3, 31.2, 29.0, 28.0, 24.7, 24.2, 22.6.

Benzyl 8-((2-(2,6-dioxopiperidin-3-yl)-1-oxoisoindolin-4-yl)amino)-8-oxoheptanoate (**6b**) (white solid, yield 85%): ^1^H NMR (400 MHz, DMSO-*d*_6_) δ 11.01 (s, 1H), 9.75 (s, 1H), 7.81 (d, *J* = 7.0 Hz, 1H), 7.49 (q, *J* = 7.4 Hz, 2H), 7.41–7.28 (m, 5H), 5.14 (dd, *J* = 13.2, 5.0 Hz, 1H), 5.08 (s, 2H), 4.43–4.29 (m, 2H), 3.00–2.83(m, 1H), 2.60 (d, *J* = 16.8 Hz, 1H), 2.35 (dd, *J* = 13.8, 6.9 Hz, 5H), 2.08–1.95 (m, 1H), 1.68–1.50 (m, 4H), 1.31 (s, 4H). ^13^C NMR (100 MHz, DMSO-*d*_6_) δ 172.8, 172.7, 171.3, 171.0, 167.8, 136.3, 133.8, 133.7, 132.6, 128.6, 128.4, 127.9, 127.9, 125.2, 118.9, 65.3, 51.5, 46.5, 40.1, 39.9, 39.7, 39.5, 39.3, 39.1, 38.9, 35.7, 33.4, 31.2, 28.3, 28.1, 24.9, 24.3, 22.6.

Benzyl 9-((2-(2,6-dioxopiperidin-3-yl)-1-oxoisoindolin-4-yl)amino)-9-oxoheptanoate (**6c**) (white solid, yield 81%): ^1^H NMR (400 MHz, DMSO-*d*_6_) δ 11.01 (s, 1H), 9.75 (s, 1H), 7.81 (d, *J* = 6.8 Hz, 1H), 7.49 (q, *J* = 7.4 Hz, 2H), 7.42–7.27 (m, 5H), 5.14 (dd, *J* = 13.2, 5.0 Hz, 1H), 5.08 (s, 2H), 4.48–4.27 (m, 2H), 3.07–2.82 (m, 1H), 2.61 (d, *J* = 17.1 Hz, 1H), 2.35 (t, *J* = 7.2 Hz, 5H), 2.10–1.94 (m, 1H), 1.56 (dd, *J* = 15.8, 8.7 Hz, 4H), 1.29 (s, 6H). ^13^C NMR (100 MHz, DMSO-*d*_6_) δ 172.8, 172.8, 171.3, 171.0, 167.8, 136.3, 133.8, 133.7, 132.6, 128.6, 128.4, 127.9, 127.9, 125.2, 118.9, 65.3, 51.5, 46.4, 35.7, 33.4, 31.2, 28.5, 28.4, 28.3, 25.0, 24.4, 22.6.

Benzyl 10-((2-(2,6-dioxopiperidin-3-yl)-1-oxoisoindolin-4-yl)amino)-10-oxoheptanoate (**6d**) (white solid, yield 75%): ^1^H NMR (400 MHz, DMSO-*d*_6_) δ 11.02 (s, 1H), 9.75 (s, 1H), 7.82 (d, *J* = 7.0 Hz, 1H), 7.49 (q, *J* = 7.3 Hz, 2H), 7.41–7.27 (m, 5H), 5.15 (dd, *J* = 13.2, 4.8 Hz, 1H), 5.08 (s, 2H), 4.45–4.28 (m, 2H), 2.93 (dd, *J* = 21.6, 8.7 Hz, 1H), 2.61 (d, *J* = 16.6 Hz, 1H), 2.34 (t, *J* = 7.1 Hz, 5H), 2.10–1.97 (m, 1H), 1.57 (dd, *J* = 16.3, 7.0 Hz, 4H), 1.26 (s, 8H). ^13^C NMR (100 MHz, DMSO-*d*_6_) δ 172.9, 172.8, 171.4, 171.0, 167.8, 136.3, 133.8, 133.7, 132.6, 128.6, 128.4, 127.9, 127.9, 125.2, 119.0, 65.2, 51.5, 46.5, 35.8, 33.4, 31.2, 28.6, 28.6, 28.5, 28.3, 25.0, 24.4, 22.6.

Benzyl 8-((2-(2,6-dioxopiperidin-3-yl)-1,3-dioxoisoindolin-4-yl)amino)-8-oxooctanoate (**6e**) (white solid, yield 64%): ^1^H NMR (400 MHz, DMSO-*d*_6_) δ 11.15 (s, 1H), 9.69 (s, 1H), 8.47 (d, *J* = 8.4 Hz, 1H), 7.83 (t, *J* = 7.9 Hz, 1H), 7.61 (d, *J* = 7.2 Hz, 1H), 7.46–7.21 (m, 5H), 5.15 (dd, *J* = 12.7, 5.3 Hz, 1H), 5.08 (s, 2H), 2.98–2.82 (m, 1H), 2.58 (dd, *J* = 22.6, 10.8 Hz, 2H), 2.45 (t, *J* = 7.4 Hz, 2H), 2.36 (t, *J* = 7.3 Hz, 2H), 2.12–2.01 (m, 1H), 1.66–1.49 (m, 4H), 1.32 (s, 4H). ^13^C NMR (100 MHz, DMSO-*d*_6_) δ 172.7, 172.7, 171.9, 169.7, 167.7, 166.6, 136.5, 136.3, 136.1, 131.4, 128.4, 127.9, 127.9, 126.3, 118.3, 117.0, 65.3, 48.9, 36.4, 33.4, 30.9, 28.1, 24.6, 24.3, 21.9.

*General procedures for the preparation of compounds **7a**–**7e**.* To a solution of **6a**–**6e** (1.0 eq) in MeOH, 10% Pd/C (10% eq) was added. The reaction mixture was purged with N_2_ and then H_2_ and stirred at room temperature for 2 h. After the reaction completed, the residue was filtered through celite and washed with DCM. The combined organic layers were then washed with saturated brine, dried over anhydrous Na_2_SO_4_, and concentrated in vacuo to produce **7a**–**7e**.

7-((2-(2,6-Dioxopiperidin-3-yl)-1-oxoisoindolin-4-yl)amino)-7-oxoheptanoic acid (**7a**) (white solid, yield 95%): ^1^H NMR (400 MHz, DMSO-*d*_6_) δ 11.03 (s, 1H), 9.88 (s, 1H), 7.82 (d, *J* = 6.1 Hz, 1H), 7.49 (d, *J* = 6.8 Hz, 2H), 5.13 (dd, *J* = 12.9, 4.3 Hz, 1H), 4.38 (q, *J* = 17.5 Hz, 2H), 2.99–2.82 (m, 1H), 2.61 (d, *J* = 16.7 Hz, 1H), 2.44–2.27 (m, 3H), 2.25–2.12 (m, 2H), 2.10–1.96 (m, 1H), 1.67–1.45 (m, 4H), 1.33 (d, *J* = 6.4 Hz, 2H). ^13^C NMR (100 MHz, DMSO-*d*_6_) δ 174.8, 172.8, 171.3, 171.0, 167.8, 133.8, 133.7, 132.6, 128.5, 125.3, 118.9, 51.5, 46.6, 40.1, 39.9, 39.7, 39.5, 39.3, 39.1, 38.9, 35.6, 34.0, 31.2, 28.2, 24.8, 24.4, 22.6, −15.0.

8-((2-(2,6-Dioxopiperidin-3-yl)-1-oxoisoindolin-4-yl)amino)-8-oxoheptanoic acid (**7b**) (white solid, yield 93%): ^1^H NMR (400 MHz, DMSO-*d*_6_) δ 11.96 (s, 1H), 11.01 (s, 1H), 9.77 (s, 1H), 7.81 (d, *J* = 6.9 Hz, 1H), 7.49 (q, *J* = 7.2 Hz, 2H), 5.14 (dd, *J* = 13.2, 5.0 Hz, 1H), 4.36 (q, *J* = 17.5 Hz, 2H), 3.04–2.82 (m, 1H), 2.61 (d, *J* = 16.0 Hz, 1H), 2.43–2.28 (m, 3H), 2.20 (t, *J* = 7.3 Hz, 2H), 2.09–1.95 (m, 1H), 1.68–1.44 (m, 4H), 1.31 (s, 4H). ^13^C NMR (100 MHz, DMSO-*d*_6_) δ 174.5, 172.8, 171.3, 171.0, 167.8, 133.7, 133.7, 132.6, 128.6, 125.2, 118.9, 51.5, 46.5, 35.7, 33.6, 31.2, 28.3, 28.3, 24.9, 24.4, 22.6.

9-((2-(2,6-Dioxopiperidin-3-yl)-1-oxoisoindolin-4-yl)amino)-9-oxoheptanoic acid (**7c**) (white solid, yield 96%): ^1^H NMR (400 MHz, DMSO-*d*_6_) δ 11.97 (s, 1H), 11.02 (s, 1H), 9.76 (s, 1H), 7.81 (dd, *J* = 6.9, 1.1 Hz, 1H), 7.49 (q, *J* = 7.2 Hz, 2H), 5.14 (dd, *J* = 13.3, 5.0 Hz, 1H), 4.36 (q, *J* = 17.5 Hz, 2H), 3.02–2.81 (m, 1H), 2.61 (d, *J* = 16.6 Hz, 1H), 2.44–2.28 (m, 3H), 2.19 (t, *J* = 7.3 Hz, 2H), 2.09–1.98 (m, 1H), 1.67–1.43 (m, 4H), 1.30 (s, 6H). ^13^C NMR (100 MHz, DMSO-*d*_6_) δ 174.5, 172.8, 171.3, 171.0, 167.8, 133.8, 133.7, 132.6, 128.6, 125.2, 118.9, 51.5, 46.4, 35.7, 33.6, 31.2, 28.5, 28.5, 28.4, 25.0, 24.4, 22.6.

10-((2-(2,6-Dioxopiperidin-3-yl)-1-oxoisoindolin-4-yl)amino)-10-oxoheptanoic acid (**7d**) (white solid, yield 90%): ^1^H NMR (400 MHz, DMSO-*d*_6_) δ 11.91 (s, 1H), 11.02 (s, 1H), 9.75 (s, 1H), 7.81 (d, *J* = 6.9 Hz, 1H), 7.54–7.43 (m, 2H), 5.15 (dd, *J* = 13.2, 5.0 Hz, 1H), 4.36 (q, *J* = 17.5 Hz, 2H), 2.99–2.82 (m, 1H), 2.61 (d, *J* = 17.3 Hz, 1H), 2.44–2.26 (m, 3H), 2.19 (t, *J* = 7.3 Hz, 2H), 2.08–1.95 (m, 1H), 1.54 (dd, *J* = 45.7, 6.5 Hz, 4H), 1.27 (d, *J* = 3.7 Hz, 8H). ^13^C NMR (100 MHz, DMSO-*d*_6_) δ 174.5, 172.8, 171.3, 171.0, 167.8, 133.8, 133.7, 132.6, 128.6, 125.2, 118.9, 51.5, 46.4, 35.8, 33.6, 31.2, 28.6, 28.5, 25.0, 24.4, 22.6.

8-((2-(2,6-Dioxopiperidin-3-yl)-1,3-dioxoisoindolin-4-yl)amino)-8-oxooctanoic acid (**7e**) (white solid, yield 89%): ^1^H NMR (400 MHz, DMSO-*d*_6_) δ 12.14 (s, 1H), 11.15 (s, 1H), 9.70 (s, 1H), 8.47 (d, *J* = 8.4 Hz, 1H), 7.83 (t, *J* = 7.8 Hz, 1H), 7.61 (d, *J* = 7.2 Hz, 1H), 5.15 (dd, *J* = 12.6, 5.3 Hz, 1H), 2.99–2.82 (m, 1H), 2.58 (dd, *J* = 23.3, 11.1 Hz, 2H), 2.46 (t, *J* = 7.4 Hz, 2H), 2.19 (t, *J* = 7.2 Hz, 2H), 2.13–2.00 (m, 1H), 1.69–1.42 (m, 4H), 1.32 (s, 4H). ^13^C NMR (100 MHz, DMSO-*d*_6_) δ 174.5, 172.7, 172.0, 169.8, 167.6, 166.6, 136.5, 136.1, 131.4, 126.3, 118.25, 117.0, 48.9, 36.4, 33.7, 30.9, 28.3, 28.2, 24.6, 24.4, 21.9.

*General procedures for the preparation of compounds **8a**–**8e**.* To a solution of **7a**–**7e** (1.0 eq) in anhydrous DMF, HATU (2.0 eq) and DIEA (3.0 eq) were added. After stirring in an ice bath for 0.5 h, O-tritylhydroxylamine (1.0 eq) was added. The mixture was stirred at room temperature overnight. After the reaction completed, the residue was diluted with H_2_O and extracted with EtOAc. The combined organic layers were then washed saturated brine, dried over anhydrous Na_2_SO_4_, and concentrated in vacuo to produce the crude product, which was purified by column chromatography with DCM/ MeOH (30:1–20:1) to produce **8a**–**8e**.

N^1^-(2-(2,6-Dioxopiperidin-3-yl)-1-oxoisoindolin-4-yl)-N^7^-(trityloxy)heptanediamide (**8a**) (white solid, yield 73%): ^1^H NMR (400 MHz, DMSO-*d*_6_) δ 11.02 (s, 1H), 10.17 (s, 1H), 9.73 (s, 1H), 7.81 (d, *J* = 6.5 Hz, 1H), 7.57–7.44 (m, 2H), 7.32 (s, 15H), 5.14 (dd, *J* = 13.1, 4.6 Hz, 1H), 4.45–4.27 (m, 2H), 2.99–2.81 (m, 1H), 2.60 (d, *J* = 17.0 Hz, 1H), 2.41–2.18 (m, 3H), 2.08–1.94 (m, 1H), 1.80 (s, 2H), 1.47 (s, 2H), 1.24 (s, 5H), 1.04 (s, 2H). ^13^C NMR (100 MHz, DMSO-*d*_6_) δ 172.8, 171.2, 171.0, 170.2, 167.8, 142.4, 133.8, 133.6, 132.6, 128.9, 128.6, 127.5, 127.4, 125.2, 118.9, 91.7, 51.5, 46.5, 35.6, 31.8, 31.2, 28.0, 24.7, 24.5, 22.6.

N^1^-(2-(2,6-Dioxopiperidin-3-yl)-1-oxoisoindolin-4-yl)-N^8^-(trityloxy)octanediamide (**8b**) (white solid, yield 79%): ^1^H NMR (400 MHz, DMSO-*d*_6_) δ 11.01 (s, 1H), 10.15 (s, 1H), 9.73 (s, 1H), 7.82 (d, *J* = 7.0 Hz, 1H), 7.49 (q, *J* = 7.4 Hz, 2H), 7.32 (s, 15H), 5.14 (dd, *J* = 13.2, 4.9 Hz, 1H), 4.46–4.28 (m, 2H), 3.00–2.83 (m, 1H), 2.60 (d, *J* = 17.3 Hz, 1H), 2.34 (dt, *J* = 14.7, 8.3 Hz, 3H), 2.09–1.95 (m, 1H), 1.78 (s, 2H), 1.60–1.41 (m, 2H), 1.19 (d, *J* = 5.8 Hz, 4H), 1.02 (d, *J* = 5.8 Hz, 2H). ^13^C NMR (100 MHz, DMSO-*d*_6_) δ 172.8, 171.3, 171.0, 170.3, 167.8, 142.4, 133.8, 133.6, 132.6, 128.9, 128.6, 127.5, 127.4, 125.2, 119.0, 91.7, 64.9, 51.5, 46.5, 40.1, 39.9, 39.7, 39.5, 39.3, 39.1, 38.9, 35.7, 31.9, 31.2, 28.3, 28.1, 24.9, 24.6, 22.6, 15.1.

N^1^-(2-(2,6-Dioxopiperidin-3-yl)-1-oxoisoindolin-4-yl)-N^9^-(trityloxy)nonanediamide (**8c**) (white solid, yield 75%): ^1^H NMR (400 MHz, DMSO-*d*_6_) δ 11.02 (s, 1H), 10.14 (s, 1H), 9.75 (s, 1H), 7.82 (d, *J* = 7.0 Hz, 1H), 7.49 (q, *J* = 7.3 Hz, 2H), 7.32 (s, 15H), 5.14 (dd, *J* = 13.2, 5.0 Hz, 1H), 4.44–4.21 (m, 2H), 2.97–2.84 (m, 1H), 2.60 (d, *J* = 18.3 Hz, 1H), 2.42–2.28 (m, 3H), 2.07–1.94 (m, 1H), 1.77 (s, 2H), 1.63–1.50 (m, 2H), 1.20 (dd, *J* = 16.3, 9.7 Hz, 6H), 1.03–0.92 (m, 2H). ^13^C NMR (100 MHz, DMSO-*d*_6_) δ 172.8, 171.3, 171.0, 170.4, 167.8, 142.4, 133.8, 133.6, 132.6, 128.9, 128.6, 127.5, 127.4, 125.2, 118.9, 91.7, 51.5, 46.4, 35.8, 32.0, 31.2, 28.5, 28.2, 26.3, 25.0, 24.7, 22.6.

N^1^-(2-(2,6-Dioxopiperidin-3-yl)-1-oxoisoindolin-4-yl)-N^10^-(trityloxy)decanediamide (**8d**) (white solid, yield 74%): ^1^H NMR (400 MHz, DMSO-*d*_6_) δ 11.02 (s, 1H), 10.13 (s, 1H), 9.75 (s, 1H), 7.82 (d, *J* = 6.5 Hz, 1H), 7.49 (q, *J* = 7.5 Hz, 2H), 7.32 (s, 15H), 5.14 (dd, *J* = 13.2, 5.0 Hz, 1H), 4.45–4.28 (m, 2H), 2.99–2.84 (m, 1H), 2.60 (d, *J* = 18.4 Hz, 1H), 2.34 (t, *J* = 7.2 Hz, 3H), 2.07–1.95 (m, 1H), 1.77 (s, 2H), 1.65–1.54 (m, 2H), 1.23 (dd, *J* = 20.0, 7.1 Hz, 8H), 0.96 (dd, *J* = 12.2, 6.4 Hz, 2H). ^13^C NMR (100 MHz, DMSO-*d*_6_) δ 172.8, 171.4, 171.0, 170.4, 167.8, 142.4, 133.8, 133.6, 132.6, 128.9, 128.6, 127.5, 127.4, 125.2, 118.9, 91.7, 51.5, 46.4, 35.8, 32.0, 31.2, 28.6, 28.3, 25.1, 24.7, 22.60, 18.8.

N^1^-(2-(2,6-Dioxopiperidin-3-yl)-1,3-dioxoisoindolin-4-yl)-N^8^-(trityloxy)octanediamide (**8e**) (white solid, yield 62%): ^1^H NMR (400 MHz, DMSO-*d*_6_) δ 11.15 (s, 1H), 10.16 (s, 1H), 9.68 (s, 1H), 8.48 (d, *J* = 8.4 Hz, 1H), 7.83 (t, *J* = 7.9 Hz, 1H), 7.62 (d, *J* = 7.3 Hz, 1H), 7.33 (s, 15H), 5.15 (dd, *J* = 12.6, 5.3 Hz, 1H), 2.97–2.82 (m, 1H), 2.58 (dd, *J* = 22.7, 10.9 Hz, 2H), 2.41 (t, *J* = 7.4 Hz, 2H), 2.12–2.03 (m, 1H), 1.78 (t, *J* = 6.6 Hz, 2H), 1.53 (dt, *J* = 14.6, 7.4 Hz, 2H), 1.19 (s, 4H), 1.05–0.94 (m, 2H). ^13^C NMR (100 MHz, DMSO-*d*_6_) δ 172.7, 172.0, 169.8, 167.7, 166.6, 142.4, 136.5, 136.1, 131.4, 128.9, 127.5, 127.4, 126.2, 118.3, 116.9, 91.5, 48.9, 48.6, 36.5, 30.9, 28.2, 28.0, 24.6, 21.9.

*General procedures for the preparation of compounds **9a**–**9e**.* To a solution of **8a**–**8e** (200 mg) in TFA (0.3 mL) and DCM (6 mL), triethylsilane (0.15·mL) was added dropwise. The mixture was stirred at room temperature for 30 min. After the reaction completed, the mixture was concentrated in vacuo to create the crude product, which was purified by column chromatography with DCM/ MeOH (20:1–15:1) to produce **9a**–**9e**.

N^1^-(2-(2,6-Dioxopiperidin-3-yl)-1-oxoisoindolin-4-yl)-N^7^-hydroxyheptanediamide (**9a**) (white solid, yield 83%): ^1^H NMR (400 MHz, DMSO-*d*_6_) δ 11.02 (s, 1H), 10.37 (s, 1H), 9.82 (s, 1H), 8.69 (s, 1H), 7.82 (d, *J* = 6.6 Hz, 1H), 7.58–7.41 (m, 2H), 5.14 (dd, *J* = 13.2, 4.9 Hz, 1H), 4.37 (q, *J* = 17.6 Hz, 2H), 3.01–2.82 (m, 1H), 2.61 (d, *J* = 17.6 Hz, 1H), 2.35 (t, *J* = 7.3 Hz, 3H), 2.08–1.89 (m, 3H), 1.55 (ddd, *J* = 29.6, 14.6, 7.3 Hz, 4H), 1.36–1.26 (m, 2H). ^13^C NMR (100 MHz, DMSO-*d*_6_) δ 172.9, 171.3, 171.1, 169.1, 167.8, 133.8, 133.7, 132.6, 128.6, 125.2, 118.9, 51.5, 46.5, 35.6, 34.2, 32.1, 31.2, 28.2, 26.3, 24.9, 24.8, 22.6. HRMS (ESI) *m*/*z* calculated for C_20_H_24_N_4_O_6_Na (M + Na)^+^: 439.1607, found: 439.1588. AlogP: 0.84; H-bond acceptor: 6; H-bond donor: 4; rotatable bonds: 8.

N^1^-(2-(2,6-Dioxopiperidin-3-yl)-1-oxoisoindolin-4-yl)-N^8^-hydroxyoctanediamide (**9b**) (white solid, yield 82%): ^1^H NMR (400 MHz, DMSO-*d*_6_) δ 11.02 (s, 1H), 10.33 (s, 1H), 9.77 (s, 1H), 8.65 (s, 1H), 7.81 (d, *J* = 6.7 Hz, 1H), 7.59–7.39 (m, 2H), 5.15 (dd, *J* = 13.2, 5.0 Hz, 1H), 4.37 (q, *J* = 17.5 Hz, 2H), 3.01–2.82 (m, 1H), 2.61 (d, *J* = 17.1 Hz, 1H), 2.35 (t, *J* = 7.3 Hz, 3H), 2.09–1.98 (m, 1H), 1.95 (t, *J* = 7.3 Hz, 2H), 1.66–1.41 (m, 4H), 1.29 (s, 4H). ^13^C NMR (100 MHz, DMSO-*d*_6_) δ 172.8, 171.3, 171.0, 169.0, 167.7, 133.8, 133.7, 132.6, 128.6, 125.2, 118.9, 51.5, 46.5, 35.7, 32.2, 31.2, 28.4, 28.3, 25.0, 24.9, 22.6. HRMS (ESI) *m*/*z* calculated for C_21_H_26_N_4_O_6_Na (M + Na)^+^: 453.1762, found: 453.1745. AlogP: 1.23; H-bond acceptor: 6; H-bond donor: 4; rotatable bonds: 9.

N^1^-(2-(2,6-Dioxopiperidin-3-yl)-1-oxoisoindolin-4-yl)-N^9^-hydroxynonanediamide (**9c**) (white solid, yield 71%): ^1^H NMR (400 MHz, DMSO-*d*_6_) δ 11.01 (s, 1H), 10.32 (s, 1H), 9.76 (s, 1H), 8.64 (s, 1H), 7.81 (d, *J* = 6.9 Hz, 1H), 7.57–7.41 (m, 2H), 5.14 (dd, *J* = 13.3, 5.0 Hz, 1H), 4.36 (q, *J* = 17.5 Hz, 2H), 3.03–2.82 (m, 1H), 2.61 (d, *J* = 16.9 Hz, 1H), 2.43–2.28 (m, 3H), 2.09–1.99 (m, 1H), 1.94 (t, *J* = 7.3 Hz, 2H), 1.51 (dd, *J* = 29.9, 22.9 Hz, 4H), 1.30 (s, 6H). ^13^C NMR (100 MHz, DMSO-*d*_6_) δ 172.8, 171.3, 171.1, 169.1, 167.8, 133.8, 133.7, 132.6, 128.6, 125.2, 118.9, 51.5, 46.5, 35.8, 32.2, 31.2, 28.5, 28.5, 28.4, 25.1, 25.0, 22.6. HRMS (ESI) *m*/*z* calculated for C_22_H_28_N_4_O_6_Na (M + Na)^+^: 467.1913, found: 467.1901. AlogP: 1.62; H-bond acceptor: 6; H-bond donor: 4; rotatable bonds: 10.

N^1^-(2-(2,6-Dioxopiperidin-3-yl)-1-oxoisoindolin-4-yl)-N^10^-hydroxydecanediamide (**9d**) (white solid, yield 84%): ^1^H NMR (400 MHz, DMSO-*d*_6_) δ 11.00 (s, 1H), 10.31 (s, 1H), 9.75 (s, 1H), 8.63 (s, 1H), 7.81 (d, *J* = 6.6 Hz, 1H), 7.50 (t, *J* = 6.7 Hz, 2H), 5.14 (dd, *J* = 13.2, 4.8 Hz, 1H), 4.36 (q, *J* = 17.4 Hz, 2H), 3.01–2.84 (m, 1H), 2.61 (d, *J* = 17.0 Hz, 1H), 2.41–2.29 (m, 3H), 2.09–1.99 (m, 1H), 1.93 (t, *J* = 7.2 Hz, 2H), 1.63–1.44 (m, 4H), 1.29 (s, 8H). ^13^C NMR (100 MHz, DMSO-*d*_6_) δ 172.9, 171.4, 171.0, 169.1, 167.8, 133.8, 133.7, 132.6, 128.6, 125.2, 118.9, 51.5, 46.4, 35.8, 32.2, 31.2, 28.7, 28.6, 28.5, 25.1, 22.6. HRMS (ESI) *m*/*z* calculated for C_23_H_30_N_4_O_6_Na (M + Na)^+^: 481.2079, found: 481.2058. AlogP: 2.01; H-bond acceptor: 6; H-bond donor: 4; rotatable bonds: 11.

N^1^-(2-(2,6-Dioxopiperidin-3-yl)-1,3-dioxoisoindolin-4-yl)-N^8^-hydroxyoctanediamide (**9e**) (white solid, yield 84%): ^1^H NMR (400 MHz, DMSO-*d*_6_) δ 11.14 (s, 1H), 10.32 (s, 1H), 9.69 (s, 1H), 8.65 (s, 1H), 8.47 (d, *J* = 8.3 Hz, 1H), 7.83 (t, *J* = 7.7 Hz, 1H), 7.61 (d, *J* = 7.1 Hz, 1H), 5.15 (dd, *J* = 12.6, 5.1 Hz, 1H), 2.99–2.80 (m, 1H), 2.58 (dd, *J* = 24.1, 10.9 Hz, 2H), 2.46 (t, *J* = 7.4 Hz, 2H), 2.11–2.01 (m, 1H), 1.94 (t, *J* = 7.1 Hz, 2H), 1.67–1.54 (m, 2H), 1.54–1.42 (m, 2H), 1.38–1.19 (m, 4H). ^13^C NMR (100 MHz, DMSO-*d*_6_) δ 172.7, 172.0, 169.8, 169.1, 167.7, 166.6, 136.5, 136.1, 131.4, 126.3, 118.3, 117.0, 48.9, 48.6, 36.5, 32.2, 30.9, 28.3, 28.2, 25.0, 24.7, 21.9. HRMS (ESI) *m*/*z* calculated for C_21_H_24_N_4_O_7_Na (M + Na)^+^: 467.1566, found: 467.1537. AlogP: 0.87; H-bond acceptor: 7; H-bond donor: 4; rotatable bonds: 9.

*General procedures for the preparation of compounds **11a**–**11b**.* To a solution of **5a** or **5b** (1.0 eq) in THF, 10 (1.5 eq) was added. The mixture was stirred at reflux for 4 h. After the reaction completed, the mixture was cooled to room temperature, filtered and washed with THF to produce **11a**–**11b**.

4-Nitrophenyl (2-(2,6-dioxopiperidin-3-yl)-1-oxoisoindolin-4-yl)carbamate (**11a**) (yellow solid, yield 74%): ^1^H NMR (400 MHz, DMSO-*d*_6_) δ 11.02 (s, 1H), 10.39 (s, 1H), 8.33 (d, *J* = 9.0 Hz, 2H), 7.81 (dd, *J* = 6.5, 2.0 Hz, 1H), 7.63–7.48 (m, 4H), 5.16 (dd, *J* = 13.2, 5.0 Hz, 1H), 4.49 (dd, *J* = 37.2, 17.7 Hz, 2H), 3.04–2.83 (m, 1H), 2.63 (d, *J* = 17.0 Hz, 1H), 2.46–2.29 (m, 1H), 2.13–1.96 (m, 1H).

4-Nitrophenyl (2-(2,6-dioxopiperidin-3-yl)-1,3-dioxoisoindolin-4-yl)carbamate (**11b**) (white solid, yield 69%): ^1^H NMR (400 MHz, DMSO-*d*_6_) δ 11.01 (s, 1H), 10.28 (s, 1H), 8.17 (d, *J* = 5.3 Hz, 2H), 7.51 (dd, *J* = 5.5, 1.7 Hz, 1H), 7.43–7.32 (m, 4H), 5.12 (d, *J* = 5.0 Hz, 1H), 3.14–2.98 (m, 1H), 2.59 (d, *J* = 12.0 Hz, 1H), 2.41–2.23 (m, 1H), 2.10–1.95 (m, 1H).

*General procedures for the preparation of compounds **13a**–**13d**.* To a solution of **11a** or **11b** (1.0 eq) in DMF, **12a**–**12c** (1.5 eq) was added. The mixture was stirred at room temperature overnight. After the reaction completed, was diluted with H_2_O and extracted with EtOAc. The combined organic layers were then washed saturated brine, dried over anhydrous Na_2_SO_4_, and concentrated in vacuo to create the crude product, which was purified by column chromatography with DCM/ MeOH (15:1–10:1) to produce **13a**–**13d**.

6-(3-(2-(2,6-Dioxopiperidin-3-yl)-1-oxoisoindolin-4-yl)ureido)hexanoic acid (**13a**) (white solid, yield 74%): ^1^H NMR (400 MHz, DMSO-*d*_6_) δ 11.02 (s, 1H), 8.68 (s, 1H), 8.04 (d, *J* = 7.9 Hz, 1H), 7.39 (t, *J* = 7.7 Hz, 1H), 7.30 (d, *J* = 7.3 Hz, 1H), 6.74 (s, 1H), 5.13 (dd, *J* = 13.1, 4.9 Hz, 1H), 4.32 (dd, *J* = 53.0, 17.1 Hz, 2H), 3.08 (d, *J* = 5.8 Hz, 2H), 2.99–2.84 (m, 1H), 2.61 (d, *J* = 16.7 Hz, 1H), 2.32 (dd, *J* = 21.9, 12.8 Hz, 1H), 2.07 (t, *J* = 7.1 Hz, 3H), 1.46 (ddd, *J* = 30.3, 13.0, 5.7 Hz, 4H), 1.33–1.22 (m, 2H). ^13^C NMR (100 MHz, DMSO-*d*_6_) δ 172.84 171.1, 168.2, 155.0, 135.8, 132.2, 130.4, 128.6, 121.5, 116.0, 51.5, 46.1, 36.3, 31.2, 29.6, 26.3, 25.1, 22.7.

7-(3-(2-(2,6-Dioxopiperidin-3-yl)-1-oxoisoindolin-4-yl)ureido)heptanoic acid (**13b**) (white solid, yield 81%): ^1^H NMR (400 MHz, DMSO-*d*_6_) δ 11.03 (s, 1H), 8.38 (s, 1H), 8.01 (t, *J* = 8.2 Hz, 1H), 7.45–7.38 (m, 1H), 7.32 (d, *J* = 7.3 Hz, 1H), 6.46 (s, 1H), 5.14 (dd, *J* = 13.2, 5.1 Hz, 1H), 4.31 (dd, *J* = 47.4, 17.0 Hz, 2H), 3.09 (dd, *J* = 12.3, 6.3 Hz, 2H), 2.99–2.86 (m, 1H), 2.61 (d, *J* = 16.3 Hz, 1H), 2.41–2.26 (m, 1H), 2.15 (t, *J* = 7.2 Hz, 1H), 2.03 (dd, *J* = 10.9, 6.0 Hz, 1H), 1.55–1.38 (m, 3H), 1.29 (s, 3H). ^13^C NMR (100 MHz, DMSO-*d*_6_) δ 172.9, 171.1, 168.1, 154.9, 135.7, 132.2, 130.5, 128.7, 123.1, 121.6, 116.1, 51.5, 45.9, 31.2, 29.5, 28.4, 26.2, 24.8, 22.7.

9-(3-(2-(2,6-Dioxopiperidin-3-yl)-1-oxoisoindolin-4-yl)ureido)nonanoic acid (**13c**) (white solid, yield 71%): ^1^H NMR (400 MHz, DMSO-*d*_6_) δ 11.95 (s, 1H), 11.03 (s, 1H), 8.27 (s, 1H), 8.03 (d, *J* = 7.8 Hz, 1H), 7.42 (dd, *J* = 16.5, 8.8 Hz, 1H), 7.32 (d, *J* = 7.3 Hz, 1H), 6.36 (s, 1H), 5.15 (dd, *J* = 13.1, 4.6 Hz, 1H), 4.30 (dd, *J* = 43.5, 16.9 Hz, 2H), 3.09 (d, *J* = 5.8 Hz, 2H), 2.93 (t, *J* = 12.9 Hz, 1H), 2.61 (d, *J* = 16.7 Hz, 1H), 2.33 (dd, *J* = 22.3, 12.9 Hz, 1H), 2.18 (t, *J* = 7.1 Hz, 2H), 2.10–1.94 (m, 1H), 1.46 (dd, *J* = 16.1, 6.3 Hz, 4H), 1.27 (s, 8H). ^13^C NMR (100 MHz, DMSO-*d*_6_) δ 174.6, 172.8, 171.1, 168.1, 154.8, 135.7, 132.3, 130.5, 128.7, 121.6, 116.1, 51.5, 45.9, 33.7, 31.2, 29.6, 28.7, 28.6, 28.5, 26.3, 24.5, 22.7.

6-(3-(2-(2,6-Dioxopiperidin-3-yl)-1,3-dioxoisoindolin-4-yl)ureido)hexanoic acid (**13d**) (white solid, yield 77%): ^1^H NMR (400 MHz, DMSO-*d*_6_) δ 12.00 (s, 1H), 11.15 (s, 1H), 8.78 (s, 1H), 8.61 (d, *J* = 8.6 Hz, 1H), 7.72 (t, *J* = 7.7 Hz, 2H), 7.42 (d, *J* = 7.1 Hz, 1H), 5.12 (dd, *J* = 12.6, 5.2 Hz, 1H), 3.10 (dd, *J* = 11.6, 5.9 Hz, 2H), 2.97–2.82 (m, 1H), 2.61 (d, *J* = 18.1 Hz, 1H), 2.22 (t, *J* = 7.2 Hz, 2H), 2.14–2.03 (m, 1H), 1.57–1.38 (m, 4H), 2.50–2.38 (m, 4H), 1.32 (dd, *J* = 14.5, 7.7 Hz, 2H). ^13^C NMR (100 MHz, DMSO-*d*_6_) δ 174.4, 172.8, 169.9, 168.3, 166.9, 154.2, 139.1, 135.8, 131.3, 124.1, 115.7, 113.5, 48.8, 33.6, 30.9, 29.0, 25.9, 24.2, 22.1.

*General procedures for the preparation of compounds **14a**–**14d**.* To a solution of **13a**–**13d** (1.0 eq) in anhydrous DMF, HATU (2.0 eq) and DIEA (3.0 eq) were added. After stirring in an ice bath for 0.5 h, O-tritylhydroxylamine (1.0 eq) was added. The mixture was stirred at room temperature overnight. After the reaction completed, the residue was diluted with H_2_O and extracted with EtOAc. The combined organic layers were then washed saturated brine, dried over anhydrous Na_2_SO_4_, and concentrated in vacuo to create the crude product, which was purified by column chromatography with DCM/ MeOH (30:1–20:1) to produce **14a**–**14d**.

6-(3-(2-(2,6-Dioxopiperidin-3-yl)-1-oxoisoindolin-4-yl)ureido)-N-(trityloxy)hexanamide (**14a**) (white solid, yield 81%): ^1^H NMR (400 MHz, DMSO-*d*_6_) δ 11.02 (s, 1H), 10.17 (s, 1H), 8.23 (s, 1H), 8.02 (d, *J* = 7.8 Hz, 1H), 7.50–7.18 (m, 16H), 6.29 (s, 1H), 5.14 (dd, *J* = 13.0, 4.3 Hz, 1H), 4.30 (dd, *J* = 41.6, 17.0 Hz, 2H), 2.96 (dd, *J* = 40.9, 9.1 Hz, 3H), 2.61 (d, *J* = 16.6 Hz, 1H), 2.32 (dd, *J* = 25.1, 15.5 Hz, 1H), 2.11–1.95 (m, 1H), 1.79 (s, 2H), 1.26 (dd, *J* = 26.0, 7.1 Hz, 4H), 1.01 (s, 2H). ^13^C NMR (100 MHz, DMSO) δ 174.4, 172.8, 169.9, 168.3, 166.9, 154.2, 139.1, 135.8, 131.3, 124.1, 115.7, 113.5, 48.8, 33.6, 30.9, 29.0, 25.9, 24.2, 22.1.

7-(3-(2-(2,6-Dioxopiperidin-3-yl)-1-oxoisoindolin-4-yl)ureido)-N-(trityloxy)heptanamide (**14b**) (white solid, yield 80%): ^1^H NMR (400 MHz, DMSO-*d*_6_) δ 11.03 (s, 1H), 10.16 (s, 1H), 8.24 (s, 1H), 8.03 (d, *J* = 7.8 Hz, 1H), 7.44–7.17 (m, 16H), 6.33 (s, 1H), 5.15 (dd, *J* = 13.1, 4.5 Hz, 1H), 4.30 (dd, *J* = 42.3, 17.0 Hz, 2H), 3.05 (d, *J* = 5.7 Hz, 2H), 2.99–2.85 (m, 1H), 2.61 (d, *J* = 16.8 Hz, 1H), 2.41–2.23 (m, 1H), 2.14–1.95 (m, 1H), 1.78 (s, 2H), 1.40–1.30 (m, 2H), 1.23–1.10 (m, 4H), 0.98 (s, 2H). ^13^C NMR (100 MHz, DMSO-*d*_6_) δ 172.8, 171.1, 170.3, 168.1, 154.8, 142.4, 135.6, 132.3, 128.9, 128.7, 127.5, 127.4, 121.7, 116.1, 91.7, 51.5, 45.9, 31.9, 31.2, 29.5, 28.1, 26.1, 24.7, 22.7.

9-(3-(2-(2,6-Dioxopiperidin-3-yl)-1-oxoisoindolin-4-yl)ureido)-N-(trityloxy)nonanamide (**14c**) (white solid, yield 88%): ^1^H NMR (400 MHz, DMSO-*d*_6_) δ 11.03 (s, 1H), 10.14 (s, 1H), 8.24 (s, 1H), 8.03 (d, *J* = 8.0 Hz, 1H), 7.47–7.17 (m, 16H), 6.33 (t, *J* = 5.2 Hz, 1H), 5.15 (dd, *J* = 13.2, 4.9 Hz, 1H), 4.30 (dd, *J* = 41.8, 16.9 Hz, 2H), 3.09 (dd, *J* = 12.3, 6.4 Hz, 2H), 3.00–2.85 (m, 1H), 2.61 (d, *J* = 15.6 Hz, 1H), 2.32 (dt, *J* = 13.1, 9.1 Hz, 1H), 2.04 (d, *J* = 5.8 Hz, 1H), 1.77 (s, 2H), 1.24 (s, 4H), 1.20–1.07 (m, 6H), 0.96 (dd, *J* = 9.2, 6.8 Hz, 2H). ^13^C NMR (100 MHz, DMSO-*d*_6_) δ 172.8, 171.1, 170.5, 168.1, 154.8, 142.4, 135.6, 132.3, 128.9, 128.7, 127.5, 127.4, 121.6, 116.0, 91.8, 51.5, 45.9, 32.0, 31.2, 29.6, 28.7, 28.6, 26.3, 24.7, 22.7.

6-(3-(2-(2,6-Dioxopiperidin-3-yl)-1,3-dioxoisoindolin-4-yl)ureido)-N-(trityloxy)hexanamide (**14d**) (white solid, yield 73%): ^1^H NMR (400 MHz, DMSO-*d*_6_) δ 11.15 (s, 1H), 10.18 (s, 1H), 8.77 (s, 1H), 8.61 (d, *J* = 8.6 Hz, 1H), 7.71 (dd, *J* = 19.2, 11.4 Hz, 2H), 7.42 (d, *J* = 7.2 Hz, 1H), 7.33 (s, 15H), 5.12 (dd, *J* = 12.7, 5.3 Hz, 1H), 3.01 (dd, *J* = 12.0, 6.2 Hz, 2H), 2.96–2.83 (m, 1H), 2.61 (d, *J* = 18.0 Hz, 1H), 2.51–2.39(m, 1H), 2.15–2.03 (m, 1H), 1.79 (t, *J* = 6.3 Hz, 2H), 1.35–1.15 (m, 4H), 1.03 (d, *J* = 6.5 Hz, 2H). ^13^C NMR (100 MHz, DMSO-*d*_6_) δ 172.8, 170.2, 169.9, 168.3, 166.8, 154.1, 142.4, 139.1, 135.8, 131.3, 128.9, 127.5, 127.4, 124.1, 115.7, 113.5, 91.7, 48.8, 31.9, 30.9, 29.0, 26.3, 25.7, 24.5, 22.1.

*General procedures for the preparation of compounds **15a**–**15d**.* To a solution of **14a**–**14d** (200 mg) in TFA (0.3 mL) and DCM (6 mL), triethylsilane (0.15 mL) was added dropwise. The mixture was stirred at room temperature for 30 min. After the reaction completed, the mixture was concentrated in vacuo to afford the crude product, which was purified by column chromatography with DCM/ MeOH (20:1–15:1) to produce **15a**–**15d**.

6-(3-(2-(2,6-Dioxopiperidin-3-yl)-1-oxoisoindolin-4-yl)ureido)-N-hydroxyhexanamide (**15a**) (white solid, yield 75%): ^1^H NMR (400 MHz, DMSO-*d*_6_) δ 10.97 (s, 1H), 10.36 (s, 1H), 8.68 (s, 1H), 8.45 (s, 1H), 8.04 (d, *J* = 7.8 Hz, 1H), 7.41 (t, *J* = 7.7 Hz, 1H), 7.32 (d, *J* = 7.3 Hz, 1H), 6.52 (s, 1H), 5.14 (dd, *J* = 13.0, 4.5 Hz, 1H), 4.31 (dd, *J* = 48.0, 17.0 Hz, 2H), 3.09 (d, *J* = 5.4 Hz, 2H), 3.00–2.84 (m, 1H), 2.62 (d, *J* = 16.8 Hz, 1H), 2.32 (dt, *J* = 13.3, 10.5 Hz, 1H), 2.09–2.00 (m, 1H), 1.96 (t, *J* = 7.0 Hz, 2H), 1.60–1.37 (m, 4H), 1.34–1.20 (m, 2H). ^13^C NMR (100 MHz, DMSO-*d*_6_) δ 172.8, 171.0, 169.0, 168.1, 154.9, 135.7, 132.2, 130.4, 128.6, 121.5, 116.1, 51.5, 45.9, 38.9, 32.2, 31.2, 29.3, 25.9, 24.9, 22.7. HRMS (ESI) *m*/*z* calculated for C_20_H_25_N_5_O_6_Na (M + Na)^+^: 454.1712, found: 454.1697. AlogP: 0.63; H-bond acceptor: 6; H-bond donor: 5; rotatable bonds: 8.

7-(3-(2-(2,6-Dioxopiperidin-3-yl)-1-oxoisoindolin-4-yl)ureido)-N-hydroxyheptanamide (**15b**) (white solid, yield 80%): ^1^H NMR (400 MHz, DMSO-*d*_6_) δ 11.01 (s, 1H), 10.35 (s, 1H), 8.66 (s, 1H), 8.35 (s, 1H), 8.03 (d, *J* = 7.7 Hz, 1H), 7.41 (t, *J* = 7.4 Hz, 1H), 7.32 (d, *J* = 7.0 Hz, 1H), 6.44 (s, 1H), 5.14 (dd, *J* = 12.9, 4.1 Hz, 1H), 4.31 (dd, *J* = 46.1, 16.9 Hz, 2H), 3.09 (d, *J* = 5.0 Hz, 2H), 3.01–2.85 (m, 1H), 2.62 (d, *J* = 17.2 Hz, 1H), 2.33 (dd, *J* = 25.8, 15.2 Hz, 1H), 2.10–2.00 (m, 1H), 1.95 (t, *J* = 6.5 Hz, 2H), 1.46 (d, *J* = 26.7 Hz, 4H), 1.27 (s, 4H). ^13^C NMR (100 MHz, DMSO-*d*_6_) δ 172.9, 171.1, 169.0, 168.1, 154.9, 135.7, 132.3, 130.5, 128.7, 121.6, 116.1, 51.5, 46.0, 32.2, 31.2, 29.5, 28.3, 26.1, 25.1, 22.7. HRMS (ESI) *m*/*z* calculated for C_21_H_27_N_5_O_6_Na (M + Na)^+^: 468.1870, found: 468.1854. AlogP: 1.03; H-bond acceptor: 6; H-bond donor: 5; rotatable bonds: 9.

9-(3-(2-(2,6-Dioxopiperidin-3-yl)-1-oxoisoindolin-4-yl)ureido)-N-hydroxynonanamide (**15c**) (white solid, yield 80%): ^1^H NMR (400 MHz, DMSO-*d*_6_) δ 11.03 (s, 1H), 10.49 (s, 1H), 8.68 (s, 1H), 8.32 (s, 1H), 8.03 (d, *J* = 7.9 Hz, 1H), 7.41 (t, *J* = 7.5 Hz, 1H), 7.32 (d, *J* = 7.3 Hz, 1H), 6.42 (s, 1H), 5.14 (d, *J* = 8.6 Hz, 1H), 4.30 (dd, *J* = 46.1, 16.9 Hz, 2H), 3.09 (d, *J* = 4.6 Hz, 2H), 3.01–2.83 (m, 1H), 2.62 (d, *J* = 16.8 Hz, 1H), 2.33 (dd, *J* = 24.7, 14.0 Hz, 1H), 2.13–2.02 (m, 1H), 1.95 (t, *J* = 6.5 Hz, 2H), 1.45 (dd, *J* = 13.4, 5.6 Hz, 4H), 1.27 (s, 8H). ^13^C NMR (100 MHz, DMSO-*d*_6_) δ 172.8, 171.1, 168.1, 154.9, 135.7, 132.3, 130.5, 128.7, 121.6, 116.1, 51.5, 45.9, 32.0, 31.2, 29.6, 28.7, 28.6, 28.5, 26.3, 25.1, 22.7. HRMS (ESI) *m*/*z* calculated for C_23_H_31_N_5_O_6_Na (M + Na)^+^: 496.2191, found: 496.2167. AlogP: 1.81; H-bond acceptor: 6; H-bond donor: 5; rotatable bonds: 11.

6-(3-(2-(2,6-Dioxopiperidin-3-yl)-1,3-dioxoisoindolin-4-yl)ureido)-N-hydroxyhexanamide (**15d**) (white solid, yield 53%): ^1^H NMR (400 MHz, DMSO-*d*_6_) δ 11.14 (s, 1H), 10.37 (s, 1H), 8.78 (s, 1H), 8.60 (d, *J* = 8.5 Hz, 1H), 7.82–7.65 (m, 2H), 7.41 (d, *J* = 7.0 Hz, 1H), 5.12 (dd, *J* = 12.6, 4.5 Hz, 1H), 3.09 (d, *J* = 4.7 Hz, 2H), 3.02–2.82 (m, 1H), 2.61 (d, *J* = 18.1 Hz, 1H), 2.20 (d, *J* = 21.8 Hz, 1H), 2.12–2.02 (m, 1H), 1.96 (t, *J* = 6.0 Hz, 2H), 1.58–1.38 (m, 4H), 1.34–1.20 (m, 2H). ^13^C NMR (100 MHz, DMSO-*d*_6_) δ 172.8, 169.9, 169.0, 168.3, 166.9, 154.2, 139.1, 135.8, 131.3, 124.1, 115.7, 113.5, 48.8, 48.5, 32.2, 30.9, 29.0, 26.0, 24.8, 22.0. HRMS (ESI) *m*/*z* calculated for C_20_H_23_N_5_O_7_Na (M + Na)^+^: 468.2683, found: 468.2672. AlogP: 0.28; H-bond acceptor: 7; H-bond donor: 5; rotatable bonds: 8.

*General procedures for the preparation of compounds **18a**–**18e**.* To a solution of **16a** or **16b** (1.0 eq) in DMF, **17** (1.1 eq) and K_2_CO_3_ (2.0 eq) were added. The mixture was stirred at 50 °C overnight. After the reaction completed, the residue was diluted with H_2_O and extracted with EtOAc. The combined organic layers were then washed saturated brine, dried over anhydrous Na_2_SO_4_, and concentrated in vacuo to create the crude product, which was purified by column chromatography with DCM/ MeOH (50:1–45:1) to produce **18a**–**18b**.

Tert-Butyl 4-(((2-(2,6-dioxopiperidin-3-yl)-1-oxoisoindolin-4-yl)oxy)methyl)benzoate (**18a**) (white solid, yield 62%): ^1^H NMR (400 MHz, DMSO-*d*_6_) δ 10.13 (s, 1H), 7.84 (d, *J* = 8.2 Hz, 2H), 7.35 (dd, *J* = 13.4, 7.9 Hz, 3H), 7.19 (d, *J* = 7.4 Hz, 1H), 7.03 (d, *J* = 7.9 Hz, 1H), 5.29 (dd, *J* = 13.4, 4.9 Hz, 1H), 4.97–4.85 (m, 2H), 4.27 (dd, *J* = 76.0, 17.1 Hz, 2H), 3.20–3.01 (m, 1H), 2.82 (d, *J* = 17.0 Hz, 1H), 2.14–2.05 (m, 1H), 1.54 (s, 9H). ^13^C NMR (100 MHz, DMSO-*d*_6_) δ 171.8, 170.6, 168.3, 164.7, 152.6, 142.3, 133.3, 130.0, 129.5, 129.0, 127.9, 127.2, 118.0, 113.8, 80.6, 52.3, 45.2, 42.6, 31.4, 27.7, 24.9, 21.6.

Tert-Butyl 4-(((2-(2,6-dioxopiperidin-3-yl)-1,3-dioxoisoindolin-4-yl)oxy)methyl)benzoate (**18b**) (white solid, yield 59%): ^1^H NMR (400 MHz, DMSO-*d*_6_) δ 11.12 (s, 1H), 7.95 (d, *J* = 8.1 Hz, 2H), 7.83 (t, *J* = 7.9 Hz, 1H), 7.63 (d, *J*= 8.1 Hz, 2H), 7.57 (d, *J* = 8.5 Hz, 1H), 7.49 (d, *J* = 7.2 Hz, 1H), 5.47 (s, 2H), 5.11 (dd, *J* = 12.9, 5.3 Hz,1H), 2.88 (d, *J* = 11.8 Hz, 1H), 2.71–2.53 (m, 2H), 2.06 (s, 1H), 1.55 (s, 9H). ^13^C NMR (100 MHz, DMSO-*d*_6_) δ 172.8, 169.9, 166.8, 165.3, 164.7, 155.2, 141.2, 137.0, 133.3, 130.8, 129.2, 126.9, 120.2, 116.7, 115.7, 80.7, 69.4, 48.8, 30.9, 27.8, 22.0.

*General procedures for the preparation of compounds **21a**–**21b**.* To a solution of **19a** or **19b** (1.0 eq) in anhydrous DMF, HATU (2.0 eq) and DIEA (3.0 eq) were added. After stirring in an ice bath for 0.5 h, **20** (1.0 eq) was added. The mixture was stirred at room temperature overnight. After the reaction completed, the residue was diluted with H_2_O and extracted with EtOAc. The combined organic layers were then washed saturated brine, dried over anhydrous Na_2_SO_4_, and concentrated in vacuo to create the crude product, which was purified by column chromatography with DCM/ MeOH (30:1–20:1) to produce **21a**–**21b**.

N-(2-Aminophenyl)-4-(((2-(2,6-dioxopiperidin-3-yl)-1-oxoisoindolin-4-yl)oxy)methyl)benzamide (**21a**) (white solid, yield 68%): ^1^H NMR (400 MHz, DMSO-*d*_6_) δ 10.98 (s, 1H), 9.66 (s, 1H), 8.00 (d, *J* = 7.7 Hz, 2H), 7.62 (d, *J* = 7.8 Hz, 2H), 7.49 (t, *J* = 7.7 Hz, 1H), 7.33 (t, *J* = 7.3 Hz, 2H), 7.17 (d, *J* = 7.5 Hz, 1H), 6.97 (t, *J* = 7.4 Hz, 1H), 6.78 (d, *J* = 7.8 Hz, 1H), 6.60 (t, *J* = 7.3 Hz, 1H), 5.36 (s, 2H), 5.12 (dd, *J* = 13.1, 4.7 Hz, 1H), 4.90 (s, 2H), 4.39 (dd, *J* = 63.0, 17.4 Hz, 2H), 2.91 (dd, *J* = 21.6, 9.3 Hz, 1H), 2.59 (d, *J* = 17.4 Hz, 1H), 2.51–2.40(m, 1H), 2.08–1.92 (m, 1H). ^13^C NMR (100 MHz, DMSO-*d*_6_) δ 172.9, 171.0, 168.0, 165.0, 153.3, 143.1, 140.0, 134.2, 133.4, 130.0, 129.8, 128.0, 127.2, 126.7, 126.5, 123.2, 116.2, 116.1, 115.4, 115.1, 68.9, 51.6, 45.1, 31.2, 22.4. HRMS (ESI) *m*/*z* calculated for C_27_H_24_N_4_O_5_Na (M + Na)**^+^**: 507.1660, found: 507.1639. AlogP: 2.86; H-bond acceptor: 6; H-bond donor: 3; rotatable bonds: 6.

N-(2-Aminophenyl)-4-(((2-(2,6-dioxopiperidin-3-yl)-1,3-dioxoisoindolin-4-yl)oxy)methyl)benzamide (**21b**) (white solid, yield 73%): ^1^H NMR (400 MHz, DMSO-*d*_6_) δ 11.11 (s, 1H), 9.67 (s, 1H), 8.02 (d, *J* = 7.9 Hz, 2H), 7.84 (t, *J* = 7.9 Hz, 1H), 7.62 (dd, *J* = 14.5, 8.3 Hz, 3H), 7.49 (d, *J* = 7.2 Hz, 1H), 7.18 (d, *J* = 7.6 Hz, 1H), 6.98 (t, *J* = 7.5 Hz, 1H), 6.79 (d, *J* = 7.9 Hz, 1H), 6.61 (t, *J* = 7.5 Hz, 1H), 5.48 (s, 2H), 5.11 (dd, *J* = 12.9, 5.3 Hz, 1H), 4.96 (s, 1H), 2.98–2.82 (m, 1H), 2.58 (dd, *J* = 19.5, 10.4 Hz, 1H), 2.11–2.01 (m, 1H). ^13^C NMR (100 MHz, DMSO-*d*_6_) δ 172., 169.9, 166.8, 165.3, 165.0, 155.3, 143.1, 139.5, 137.0, 134.2, 133.3, 128.0, 126.9, 126.7, 126.5, 123.2, 120.2, 116.7, 116.2, 116.1, 115.7, 69.5, 53.6, 48.8, 30.9, 22.0, 18.0. HRMS (ESI) *m*/*z* calculated for C_27_H_22_N_4_O_6_Na (M + Na)**^+^**: 521.1443, found: 521.1432. AlogP: 2.50; H-bond acceptor: 7; H-bond donor: 3; rotatable bonds: 6.

*General procedures for the preparation of compounds **22a**–**22b**.* To a solution of **19a** or **19b** (1.0 eq) in anhydrous DMF, HATU (2.0 eq) and DIEA (3.0 eq) were added. After stirring in an ice bath for 0.5 h, O-tritylhydroxylamine (1.0 eq) was added. The mixture was stirred at room temperature overnight. After the reaction completed, the residue was diluted with H_2_O and extracted with EtOAc. The combined organic layers were then washed with saturated brine, dried over anhydrous Na_2_SO_4_, and concentrated in vacuo to create the crude product, which was purified by column chromatography with DCM/ MeOH (50:1–40:1) to produce **22a**–**22b**.

4-(((2-(2,6-Dioxopiperidin-3-yl)-1-oxoisoindolin-4-yl)oxy)methyl)-N-(trityloxy)benzamide (**22a**) (white solid, yield 77%): ^1^H NMR (400 MHz, CDCl_3_) δ 8.45 (s, 1H), 8.24 (s, 1H), 7.49 (d, *J* = 5.9 Hz, 6H), 7.37–7.13 (m, 15H), 6.96 (d, *J* = 7.9 Hz, 1H), 5.07 (dd, *J* = 13.3, 4.8 Hz, 1H), 4.87–4.76 (m, 2H), 4.30–4.16 (m, 2H), 3.08–2.65 (m, 2H), 2.32–2.10 (m, 1H), 2.07 (dd, *J* = 5.9, 4.1 Hz, 1H).

4-(((2-(2,6-Dioxopiperidin-3-yl)-1,3-dioxoisoindolin-4-yl)oxy)methyl)-N-(trityloxy)benzamide (**22b**) (white solid, yield 72%): ^1^H NMR (400 MHz, CDCl_3_) δ 8.18 (s, 1H), 7.98 (s, 1H), 7.62 (t, *J* = 7.8 Hz, 2H), 7.51 (s, 4H), 7.43–7.28 (m, 15H), 7.15 (d, *J* = 8.4 Hz, 1H), 5.28 (s, 2H), 4.96 (dd, *J* = 12.0, 5.2 Hz, 1H), 2.80 (d, *J* = 11.0 Hz, 3H), 2.12 (dt, *J* = 10.4, 6.3 Hz, 1H).

*General procedures for the preparation of compounds **23a**–**23b**.* To a solution of **22a** or **22b** (200 mg) in TFA (0.3 mL) and DCM (6 mL), triethylsilane (0.15 mL) was added dropwise. The mixture was stirred at room temperature for 30 min. After the reaction completed, the mixture was concentrated in vacuo to create the crude product, which was purified by column chromatography with DCM/ MeOH (20:1–15:1) to produce **23a**–**23b**.

4-(((2-(2,6-Dioxopiperidin-3-yl)-1-oxoisoindolin-4-yl)oxy)methyl)-N-hydroxybenzamide (**23a**) (white solid, yield 67%): ^1^H NMR (400 MHz, DMSO-*d*_6_) δ 11.17 (s, 1H), 10.13 (s, 1H), 9.00 (s, 1H), 7.68 (d, *J* = 8.2 Hz, 2H), 7.33 (dd, *J* = 14.0, 7.8 Hz, 2H), 7.19 (d, *J* = 7.3 Hz, 1H), 7.02 (d, *J* = 7.9 Hz, 1H), 5.29 (dd, *J* = 13.3, 4.9 Hz, 1H), 4.96–4.78 (m, 2H), 4.27 (dd, *J* = 74.7, 17.1 Hz, 2H), 3.20–3.05 (m, 1H), 2.82 (d, *J* = 17.5 Hz, 1H),2.50–2.42(m, 1H), 2.14–2.03 (m, 1H). ^13^C NMR (100 MHz, DMSO-*d*_6_) δ 171.9, 170.7, 168.3, 164.0, 152.6, 140.3, 133.3, 131.5, 129.5, 128.0, 127.0, 126.9, 118.0, 113.8, 54.9, 52.3, 45.3, 42.6, 31.4, 21.7. HRMS (ESI) *m*/*z* calculated for C_21_H_19_N_3_O_6_Na (M + Na)^+^: 432.1178, found: 432.1166. AlogP: 1.15; H-bond acceptor: 6; H-bond donor: 3; rotatable bonds: 5.

4-(((2-(2,6-Dioxopiperidin-3-yl)-1,3-dioxoisoindolin-4-yl)oxy)methyl)-N-hydroxybenzamide (**23b**) (white solid, yield 71%): ^1^H NMR (400 MHz, DMSO-*d*_6_) δ 11.24 (s, 1H), 11.12 (s, 1H), 9.06 (s, 1H), 7.92–7.75 (m, 3H), 7.58 (d, *J* = 7.6 Hz, 3H), 7.49 (d, *J* = 7.0 Hz, 1H), 5.43 (s, 2H), 5.11 (dd, *J* = 12.8, 5.0 Hz, 1H), 2.96–2.83 (m, 1H), 2.67–2.53 (m, 2H), 2.11–1.98 (m, 1H). ^13^C NMR (100 MHz, DMSO-*d*_6_) δ 172.7, 169.9, 166.7, 165.3, 163.8, 155.3, 139.3, 137.0, 133.2, 132.3, 127.0, 126.9, 120.1, 116.6, 115.6, 69.4, 48.7, 30.9, 21.9. HRMS (ESI) *m*/*z* calculated for C_21_H_17_N_3_O_7_Na (M + Na)^+^: 446.2820, found: 446.2801. AlogP: 0.79; H-bond acceptor: 7; H-bond donor: 3; rotatable bonds: 5.

### 4.4. Biological Evaluation Methods

#### 4.4.1. In Vitro HDAC Inhibition Assay

All three full-length rhHDACs were expressed in insect High5 cells using a baculoviral expression system, and all the His6-tagged and GST-fusion proteins were purified using Ni-NTA (QIAGEN). The deacetylase activity of rhHDACs (recombinant human HDACs) 1 and 3 were assayed with an HDAC substrate (Ac-Lys-Tyr-Lys(ε-acetyl)-AMC), and the HDAC6 was assayed with another HDAC substrate (Boc-Lys(ε-acetyl)-AMC).The total HDAC assay volume was 25 μl and all the assay components were diluted in a Hepes buffer (25 mM Hepes, 137 mM NaCl, 2.7 mM KCl, and 4.9 mM MgCl2; pH 8.0).The reaction was carried out in black 384 well plates (OptiPlateTM-384F, PerkinElmer). In brief, the HDAC assay mixture contained an HDAC substrate (5–50 μM, 5 μl), rhHDAC isoforms (20–200 nM) and an inhibitor (1 μl). The positive controls contained all the above components except the inhibitor. The negative controls contained neither an enzyme nor an inhibitor. The HDAC6 assay components were incubated at room temperature for 3 h, and HDAC 1 and 3 were incubated for 24 h. The reaction was quenched with the addition of 25 μl trypsin (diluted to final concentration 0.3125%). The plates were incubated for 30 min at room temperature to allow the fluorescence signal to develop. The fluorescence generated was monitored at wavelengths 355 nm (excitation) and 460 nm (emission) using Envision (PerkinElmer).

#### 4.4.2. Anti-Proliferative Activities

The cells were seeded in 96 well plates at a proper density in growth media. After 24 h, a range of concentrations of the test compounds was added and the plates were incubated at 37 °C for 72 h. The cell proliferation was determined by the MTS assay. After 3 h, the absorbance was collected using SpectraMAX 340. The IC_50_ values were calculated by fitting a concentration–response curve using a SoftMax pro-based four-parameter method.

#### 4.4.3. Western Blot Analysis

The 2 × 10^6^ MM.1S cells (multiple myeloma) were cultured in a cell culture plate and incubated for 1 h. Next, the MM.1S cells were treated with compounds at different concentrations for 3 h. Subsequently, the cells were collected, washed, lysed in a Ripa buffer containing a sample reducing agent, and analyzed using SDS/PAGE. Finally, the gel was blotted and the total HDAC levels were detected through standard Western blot.

## Data Availability

Not applicable.

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
