# Peer review of "Design, Synthesis, and Biological Evaluation of HDAC Degraders with CRBN E3 Ligase Ligands"

_molecules, 2021, doi:10.3390/molecules26237241_

Round 1

Reviewer 1 Report

Dear Editor and Authors,

The article is of a very good professional level, however, in my opinion, it requires improvement in terms of evidence of the pathways of the alkylation reactions of compounds 16a and 16b with bromide 17. The fact is that this reaction can proceed in two ways: 1. according to the phenolic oxygen given in the article. 2. on the imide nitrogen of the glutarimide ring. Unfortunately, the presented physicochemical data for compounds 18a and 18b do not provide unambiguous evidence of the pathways of the processes. It is necessary for the authors to carry out additional spectral studies for unequivocal evidence of the indicated structures and the corresponding compounds obtained from them.

Author Response

The article is of a very good professional level, however, in my opinion, it requires improvement in terms of evidence of the pathways of the alkylation reactions of compounds 16a and 16b with bromide 17. The fact is that this reaction can proceed in two ways: 1. according to the phenolic oxygen given in the article. 2. on the imide nitrogen of the glutarimide ring. Unfortunately, the presented physicochemical data for compounds 18a and 18b do not provide unambiguous evidence of the pathways of the processes. It is necessary for the authors to carry out additional spectral studies for unequivocal evidence of the indicated structures and the corresponding compounds obtained from them.

Answer:

Thank you for your constructive suggestions. NMR spectra has been added in the Supporting Information part, which indicated that this reaction proceeded on the phenolic oxygen.

Reviewer 2 Report

molecules-1488850

This manuscript describes the design and synthesis of new histone deacetylase PROTACs and the evaluation of their inhibitory activity as well as their antiproliferative activity against MM.1S cells. It is a well written and interesting paper, the synthetic pathways are clearly presented, the compounds are well characterized and the biological evaluation, provides important information, even if, as the authors state, more work is needed to elucidate the exact mechanisms. The work could be published upon minor revision.

Please find below my suggestions.

Introduction, line 32 instead of “Eighteen HDACs are fell into two families”, better use “Eighteen HDACs are classified into two families”

Introduction, line 36 “dinucleotide”

Introduction, line 68 instead of “CC-220 has been designed”, better use “CC-220 (Scheme 1) has been designed”

Scheme 1, should better be defined as “Figure 3”. In this case, all Schemes that follow must be renumbered.

Results and Discussion, line 107, better use “to carbamates 11a and 11b, respectively,”

Results and Discussion, lines 119-120 instead of “the intermediates (19a~19b) were obtained by a substitution reaction followed by removal of the Boc protecting group…”, better use “compounds 19a~19b were obtained by a substitution reaction resulting in 18a-18b, followed by removal of the Boc protecting group…”

It seems to me that the paragraph “2.2. Design and Biological Evaluation” should be “2.2. Discussion and Biological Evaluation”

Design and Biological Evaluation, line 130 “plays”

Design and Biological Evaluation, line 133 instead of “Although”, better use “It should be noticed that”

Design and Biological Evaluation, line 185, it should be “degrader 21a”

Author Response

This manuscript describes the design and synthesis of new histone deacetylase PROTACs and the evaluation of their inhibitory activity as well as their antiproliferative activity against MM.1S cells. It is a well written and interesting paper, the synthetic pathways are clearly presented, the compounds are well characterized and the biological evaluation, provides important information, even if, as the authors state, more work is needed to elucidate the exact mechanisms. The work could be published upon minor revision.

Please find below my suggestions.

Introduction, line 32 instead of “Eighteen HDACs are fell into two families”, better use “Eighteen HDACs are classified into two families”

Introduction, line 36 “dinucleotide”

Introduction, line 68 instead of “CC-220 has been designed”, better use “CC-220 (Scheme 1) has been designed”

Scheme 1, should better be defined as “Figure 3”. In this case, all Schemes that follow must be renumbered.

Results and Discussion, line 107, better use “to carbamates 11a and 11b, respectively,”

Results and Discussion, lines 119-120 instead of “the intermediates (19a~19b) were obtained by a substitution reaction followed by removal of the Boc protecting group…”, better use “compounds 19a~19b were obtained by a substitution reaction resulting in 18a-18b, followed by removal of the Boc protecting group…”

It seems to me that the paragraph “2.2. Design and Biological Evaluation” should be “2.2. Discussion and Biological Evaluation”

Design and Biological Evaluation, line 130 “plays”

Design and Biological Evaluation, line 133 instead of “Although”, better use “It should be noticed that”

Design and Biological Evaluation, line 185, it should be “degrader 21a”

Answer:

Thank you for your constructive suggestions and we have revised them in the manuscript.

Reviewer 3 Report

This research work is in good agreement with the journal’s scope and limitations. I found this manuscript with excellent presentation of results and clearly describing the outcome and topic is interesting for the readers and follow the scope of the journal Molecules.

The author did a detailed study that is supported by the experimental data; however, the author needs to provide NMR spectra of all new products in the supporting information which could provide details of the purity of these compounds as well. The author needs to discuss biology a little more which lack in this article. The author has synthesized a new set of analogs with enough variables for Structure-activity relationship studies, although there is a scope for new analogs, hope the author would discuss this in his next article while addressing to improve potency, physicochemical properties, and PK profiles. 

I would like to recommend the article could be published in the journal molecules if the author would address the following queries to improve the quality of the manuscript at the next level.

  1. The Author should rewrite the sentence “To date, several hydroxamic acid and benzamide-type HDACi have been approved by the FDA (Figure 1)”: Not several HDACi have been approved only a few after hundreds of HDACi gone through a clinical trial.
  2. The author should show a general diagram depicting key HDACi interactions (e.g. Zinc binding group (ZBG), Linker, cap, and End group (Isomer selectivity). 
  3. Are these new PROTACs mutagenic in the Ames test? As hydroxamates are known for mutagenicity, any comments on this.
  4. Figure 2 should contain a general diagram depicting “anchor”; a protein of interest (POI); “bridge” etc for better understanding go key interactions.
  5. Scheme 1 should be represented as Figure 1.
  6. The author needs to provide an explanation for choosing these linkers, and some more analogs could have been designed.
  7. Any reason the authors didn’t screen any carbamates analog.
  8. Any effect of enantiomers of Lenalidomide, pomalidomide, and CC-220 moiety in the activity.
  9. Line 96: “condensation reaction” should be an amide reaction.
  10. The authors need to provide a range of yield and reaction time in the schemes (2-4) and make the reaction sequences in order.
  11. For compounds (9a-9e) author could have treated (6a-6e or its methyl analogs) with NH2OH to get the final product. A similar strategy should have been used for the other analogs as well.
  12. The author needs to provide a valid explanation for the “Unfortunately, compounds 9a and 9c showed sharply decreased potencies in the inhibition of HDAC1 and HDAC3….”.
  13. For any reason, these compounds were not screened against other HDACs.
  14. The author could have included the structure of the molecules in table 1.
  15. The author could have done other assay for validity test of the current assay. Are these results triplicated?

 “Interestingly, the HDAC inhibitory abilities of 9b, 9e, 15a, and 15d were significantly stronger than those of 21a, 21b, and 23b. The discrepancy of IC50s for the inhibition of HDAC6 and anti-proliferation for compound 21a strongly suggests that the anti-proliferation of 21a operates under a very different mechanism.:

  1. The physical and chemical properties (LogP, TPSA, pKA, HBA, HBD, rotatable bonds) of the synthesized compound are missing to show these molecules having druglike properties.
  2. In the experimental section: 

(a) The author needs to mention the amount of DMF/MeOH (mL/mmol) was used in each case.

(b) Spectra are missing for compounds (6, 7, 8, 11,12, 13, 14, 18, and 22) and the author should provide experimental data for all new compounds with spectra. Provide the missing data.

(c) For 13C, the author needs to provide only one decimal value.

(d) Extra peak at ~5.75 for compound 9c.

(e) Compounds 15 are not very pure.

(f) Compound 21b has an extra peak at ~8 .0 ppm and 13 C showing impurities: not publishable standard.

(g) For compound 23a-23b: The number of protons is less in the 1H NMR and has extra between 5.5-6.0 ppm.

(h) Explain the extra peak between 8-9 ppm for compound 23b.

Author Response

This research work is in good agreement with the journal’s scope and limitations. I found this manuscript with excellent presentation of results and clearly describing the outcome and topic is interesting for the readers and follow the scope of the journal Molecules.

The author did a detailed study that is supported by the experimental data; however, the author needs to provide NMR spectra of all new products in the supporting information which could provide details of the purity of these compounds as well. The author needs to discuss biology a little more which lack in this article. The author has synthesized a new set of analogs with enough variables for Structure-activity relationship studies, although there is a scope for new analogs, hope the author would discuss this in his next article while addressing to improve potency, physicochemical properties, and PK profiles. 

I would like to recommend the article could be published in the journal molecules if the author would address the following queries to improve the quality of the manuscript at the next level.

  1. The Author should rewrite the sentence “To date, several hydroxamic acid and benzamide-type HDACi have been approved by the FDA (Figure 1)”: Not several HDACi have been approved only a few after hundreds of HDACi gone through a clinical trial.

Answer:

Thank you for your constructive suggestions and we have revised them in the manuscript.

  1. The author should show a general diagram depicting key HDACi interactions (e.g. Zinc binding group (ZBG), Linker, cap, and End group (Isomer selectivity). 

Answer:

Thank you for your constructive suggestions and we have revised them in the manuscript.

  1. Are these new PROTACs mutagenic in the Ames test? As hydroxamates are known for mutagenicity, any comments on this.

Answer:

Thank you for your constructive suggestions and Ames test was not performed on these compounds. In this article, we mainly focused on new structures and Ames test will be performed in future evaluation.

  1. Figure 2 should contain a general diagram depicting “anchor”; a protein of interest (POI); “bridge” etc for better understanding go key interactions.

Answer:

Thank you for your constructive suggestions and we have revised them in the manuscript in Figure 2.

  1. Scheme 1 should be represented as Figure 1.

Answer:

Thank you for your constructive suggestions and we have revised them in the manuscript.

  1. The author needs to provide an explanation for choosing these linkers, and some more analogs could have been designed.

Answer:

Thank you for your constructive suggestions. The design strategies and rules of PROTACs have not been fully elucidated, but the role of the linker exerts a defining effect on physicochemical properties and biological activity of PROTAC compounds. Here, we chose these alkyl linkers with various linker length which was employed in many PROTACs to find the right combination of linkers in our PROTACs. The benzyl alcohol linkage was also employed for the first time and the main cause was to develop more types of potent PROTACs for targeting more POI.

  1. Any reason the authors didn’t screen any carbamates analog.

Answer:

Thank you for your constructive suggestions. Carbamates analogs were screened in HDAC assay and anti-proliferative assay.

  1. Any effect of enantiomers of Lenalidomide, pomalidomide, and CC-220 moiety in the activity.

Answer:

Thank you for your constructive suggestions. Enantiomers of lenalidomide, pomalidomide, and CC-220 were not employed in the design of the compounds.

  1. Line 96: “condensation reaction” should be an amide reaction.

Answer:

Thank you for your constructive suggestions and we have revised them in the manuscript.

  1. The authors need to provide a range of yield and reaction time in the schemes (2-4) and make the reaction sequences in order.

Answer:

Thank you for your constructive suggestions and we have revised them in the manuscript.

  1. For compounds (9a-9e) author could have treated (6a-6e or its methyl analogs) with NH2OH to get the final product. A similar strategy should have been used for the other analogs as well.

Answer:

Thank you for your constructive suggestions.

  1. The author needs to provide a valid explanation for the “Unfortunately, compounds 9a and 9c showed sharply decreased potencies in the inhibition of HDAC1 and HDAC3….”.

Answer:

Thank you for your constructive suggestions. The design strategies and rules of PROTACs have not been fully elucidated. The linker composition and length play a key role in potency. If the linker is too short, the two ligands cannot simultaneously bind to their respective proteins due to steric clashes. On the other hand, if the linker is too long then two proteins cannot be brought in close proximity with each other for target ubiquitination. These reasons may contribute to the decreased potencies in the inhibition of HDAC1 and HDAC3.

  1. For any reason, these compounds were not screened against other HDACs.

Answer:

Thank you for your constructive suggestions. Based on the previous research in our group, we mainly focused on HDAC1, HDAC3, and HDAC6 in this manuscript.

  1. The author could have included the structure of the molecules in table 1.

Answer:

Thank you for your constructive suggestions and we have included the structure of the molecules in Table 1.

  1. The author could have done other assay for validity test of the current assay. Are these results triplicated?

Answer:

Thank you for your constructive suggestions. These results are triplicated and all tested values were expressed as IC50 (μM ± SD).

 “Interestingly, the HDAC inhibitory abilities of 9b, 9e, 15a, and 15d were significantly stronger than those of 21a, 21b, and 23b. The discrepancy of IC50s for the inhibition of HDAC6 and anti-proliferation for compound 21a strongly suggests that the anti-proliferation of 21a operates under a very different mechanism.:

  1. The physical and chemical properties (LogP, TPSA, pKA, HBA, HBD, rotatable bonds) of the synthesized compound are missing to show these molecules having druglike properties.

Answer:

Thank you for your constructive suggestions and we have revised them in the manuscript.

  1. In the experimental section: 
  • The author needs to mention the amount of DMF/MeOH (mL/mmol) was used in each case.

(b) Spectra are missing for compounds (6, 7, 8, 11,12, 13, 14, 18, and 22) and the author should provide experimental data for all new compounds with spectra. Provide the missing data.

(c) For 13C, the author needs to provide only one decimal value.

(d) Extra peak at ~5.75 for compound 9c.

(e) Compounds 15 are not very pure.

(f) Compound 21b has an extra peak at ~8 .0 ppm and 13 C showing impurities: not publishable standard.

(g) For compound 23a-23b: The number of protons is less in the 1H NMR and has extra between 5.5-6.0 ppm.

(h) Explain the extra peak between 8-9 ppm for compound 23b.

Answer:

Thank you for your constructive suggestions and we have revised them in the manuscript and supporting information. Extra peak at ~5.75 is DCM and extra peak at ~8 .0 ppm is CDCl3. Compounds 15 were tested in HPLC and the purity is larger than 95%.